

# Native language identification from text using a fine-tuned GPT-2 model

Yuzhe Nie

School of Foreign Languages, Shanghai University, Shanghai, China

## ABSTRACT

Native language identification (NLI) is a critical task in computational linguistics, supporting applications such as personalized language learning, forensic analysis, and machine translation. This study investigates the use of a fine-tuned GPT-2 model to enhance NLI accuracy. Using the NLI-PT dataset, we preprocess and fine-tune GPT-2 to classify the native language of learners based on their Portuguese-written texts. Our approach leverages deep learning techniques, including tokenization, embedding extraction, and multi-layer transformer-based classification. Experimental results show that our fine-tuned GPT-2 model significantly outperforms traditional machine learning methods (*e.g.*, SVM, Random Forest) and other pre-trained language models (*e.g.*, BERT, RoBERTa, BioBERT), achieving a weighted F1 score of 0.9419 and an accuracy of 94.65%. These results show that large transformer models work well for native language identification and can help guide future research in personalized language tools and artificial intelligence (AI)-based education.

## INTRODUCTION

Native language identification (NLI) represents a significant challenge in the domain of computational linguistics, playing an essential role in understanding linguistic diversity and enhancing cross-cultural communication. The precise identification of a speaker's native language enables the development of customized language processing applications, thereby improving user experiences in multilingual settings. By examining phonetic, syntactic, and lexical characteristics of both spoken and written language, NLI models are capable of effectively differentiating between various languages and dialects. As global interactions increase and the volume of language data expands, the need for robust, scalable, and automated NLI systems has become increasingly critical, propelling advancements in natural language processing and machine learning. As globalization continues to blur linguistic boundaries, grasping the intricacies of NLI is vital for promoting intercultural communication and safeguarding linguistic diversity (*Baimyrza et al., 2024*; *Schroeder, Lam & Marian, 2015*).

The intricacy of NLI has intensified in recent years, influenced by factors such as rising migration rates, the expansion of digital communication, and the rise of multilingual societies. These trends require a reassessment of conventional frameworks that have traditionally governed the classification and identification of languages. The relationship

Corresponding author
Yuzhe Nie, nieyuzhe1900@163.com

between language proficiency, cultural affiliation, and identity formation further complicates the accurate determination of an individual's native language. For example, research indicates that bilingual individuals often navigate a fluid linguistic identity, which poses challenges in ascertaining their primary language (*Wang, 2016*; *Cárdenas & Verkuyten, 2019*). This complexity is further heightened by sociolinguistic factors that affect language usage in various contexts, underscoring the necessity for a more refined understanding of NLI that incorporates these elements.

The practical applications of NLI are both diverse and impactful. In educational settings, understanding a learner's native language can inform customized instructional strategies, addressing specific language transfer challenges encountered during second language acquisition (*Vajjala & Banerjee, 2017*; *Goldin, Rabinovich & Wintner, 2018*; *Mohammadi, Veisi & Amini, 2017*). Learners' errors are often reflective of their native language influences, and recognizing these patterns can aid in the development of effective teaching materials and interventions (*Lotfi, Markov & Daelemans, 2020*; *Goldin, Rabinovich & Wintner, 2018*). Additionally, NLI has significant implications in forensic linguistics, where it assists in authorship attribution and the analysis of written evidence within legal contexts (*Vajjala & Banerjee, 2017*; *Lotfi, Markov & Daelemans, 2020*; *Krebbers, Kaya & Karpov, 2022*). Identifying a writer's native language can also enhance automated language processing systems, improving the accuracy of speech recognition and machine translation technologies by tailoring them to the linguistic characteristics of specific user groups (*Sarwar et al., 2020*; *Lotfi, Markov & Daelemans, 2020*). Furthermore, NLI is increasingly pertinent in analyzing linguistic patterns in social media and user-generated content across various demographics and cultural backgrounds (*Goldin, Rabinovich & Wintner, 2018*; *Krebbers, Kaya & Karpov, 2022*). This analysis provides insights into language use in contexts such as targeted marketing strategies for specific linguistic communities and understanding multilingual communication dynamics in digital environments (*Lotfi, Markov & Daelemans, 2020*; *Krebbers, Kaya & Karpov, 2022*). Additionally, the task holds potential for security applications, where identifying individuals' native languages can aid in profiling and risk assessments in scenarios such as immigration and border control.

In modern linguistic analysis, the complexity of language transfer phenomena poses challenges in accurately an author's native language based on their written text in a second language. The influence of a speaker's first language can lead to distinct errors and stylistic choices in their second language writing, complicating the identification process (*Vajjala & Banerjee, 2017*; *Sarwar et al., 2020*; *Lotfi, Markov & Daelemans, 2020*; *Markov, Nastase & Strapparava, 2020*). As a result, the NLI task has garnered attention in computational linguistics and natural language processing, with researchers striving to create robust models capable of accurately classifying native languages based on written outputs (*Cimino & Dell'Orletta, 2017*; *Goldin, Rabinovich & Wintner, 2018*). Metadata enrichment and machine learning have emerged as powerful tools for enhancing NLI. The integration of advanced computational techniques allows researchers to analyze vast datasets, uncovering patterns and correlations that may not be readily apparent through traditional methods. Machine learning algorithms can be trained to recognize linguistic features associated with

specific languages, thereby improving the accuracy of NLI systems. Moreover, the incorporation of metadata, such as demographic information, language exposure, and cultural background, can provide valuable insights into the factors influencing language identification (*Jain, Ajay & Kumaraswamy, 2017*). This technological advancement represents a paradigm shift in linguistic research, enabling a more comprehensive exploration of the complexities surrounding NLI.

However, the integration of AI-driven approaches into traditional linguistic frameworks presents several challenges. Balancing the rigor of established linguistic theories with the flexibility of machine learning models requires careful consideration of methodological rigor and ethical implications. Researchers must navigate the potential biases inherent in algorithmic decision-making, ensuring that NLI systems are equitable and representative of diverse linguistic communities (*Zhang, 2018*; *Hierro, 2015*). Additionally, the reliance on technology raises questions about the role of human agency in language identification, as automated systems may overlook the subjective experiences of individuals regarding their linguistic identities. As such, a collaborative approach that combines traditional linguistic insights with modern technological advancements is essential for advancing the field of NLI.

NLI is a complex and multifaceted area of study that encompasses linguistic, cultural, and technological dimensions. As the world becomes increasingly interconnected, the importance of understanding and accurately identifying native languages cannot be overstated. By synthesizing traditional linguistic frameworks with modern computational techniques, researchers can contribute to a more nuanced understanding of language identification that respects the rich tapestry of human identity and cultural heritage. The ongoing exploration of NLI will undoubtedly yield valuable insights that inform not only academic discourse but also practical applications in our diverse and multilingual societies.

## Motivation

In recent years, the rapid advancement of artificial intelligence (AI) has profoundly transformed numerous industries, with education being one of the most notably impacted. Among these innovations, generative language models such as generative pre-trained transformer (GPT) have revolutionized language learning by offering interactive and personalized educational experiences. Developed by OpenAI, ChatGPT is based on GPT architecture and leverages deep learning techniques to generate human-like text in response to user input. Its ability to produce coherent, context-aware responses has attracted considerable interest in language education, where it enables learners to practice and refine their language skills through simulated dialogue and dynamic feedback (*Grassini, 2023*).

Within the field of NLI, transformer-based models like GPT offer promising avenues to improve the accuracy and robustness of language detection systems. NLI plays a critical role in applications such as machine translation, sentiment analysis, and adaptive educational technologies. By harnessing the advanced natural language processing capabilities of large language models, researchers can develop more sophisticated methods for identifying a user's native language from written text, even in complex multilingual or

code-mixed contexts (*Abinaya et al., 2023*; *Kohnke, Moorhouse & Zou, 2023*). This is particularly relevant in today's interconnected world, where linguistic diversity and fluid language use present ongoing challenges for traditional classification techniques.

Transformer models trained on extensive multilingual corpora are particularly well-suited for addressing these challenges. Unlike conventional approaches, which often struggle with dialectal variations and language mixing, models such as GPT demonstrate flexibility and adaptability in handling such complexities. This capacity is especially valuable in educational environments, where accurately identifying a learner's linguistic background can inform the design of more effective, individualized instruction (*Bhansali et al., 2022*; *Thara & Poornachandran, 2021*).

In this study, we investigate the application of a fine-tuned GPT-2 model (*Radford et al., 2019*) for NLI tasks. Our approach aims not only to classify native languages from written Portuguese texts but also to improve the scalability and accuracy of language identification across diverse linguistic inputs. By applying deep learning techniques to this task, we contribute a practical solution for enhancing language-aware technologies used in online platforms, educational applications, and information retrieval systems globally.

## RELATED WORK

The field of NLI has gained considerable traction in linguistics and natural language processing (NLP) over recent decades. Traditional investigations have predominantly centered on identifying the traits that differentiate native speakers from non-native speakers, with an emphasis on linguistic competence, heritage, and affiliation as crucial elements of native language identity (*Kozhemyakova et al., 2019*). *Goldin, Rabinovich & Wintner (2018)* established a foundation for examining how linguistic features appear in written texts, suggesting that the errors made by second language learners often reflect influences from their native languages. This initial work has been instrumental in developing methodologies for identifying native languages based on written outputs produced in a second language.

In recent years, NLI research has transitioned towards more advanced computational techniques, incorporating machine learning algorithms to enhance accuracy. The notable study of *Malmasi & Dras (2017)*, which proposed adoption of ensemble methods and deep learning architectures has notably improved the performance of NLI systems. Researchers have investigated various feature extraction methods, such as character and word n-grams, to capture the subtleties of language use that indicate a writer's native language (*Mohammadi, Veisi & Amini, 2017*). The creation of extensive datasets, like the TOEFL11 *corpus*, has equipped researchers with valuable resources for training and evaluating NLI models, thereby increasing the reliability of findings (*Blanchard et al., 2013*). *Malmasi et al. (2017)* highlighted this shift, demonstrating competitive systems that employed a variety of classifiers to achieve high accuracy in identifying the native languages of English as a second language (ESL) writers.

Despite these advancements, the NLI task still encounters several challenges. A significant obstacle is the inherent variability in language use among individuals, complicating the development of universally applicable models. The overlap of linguistic

features across various native languages can lead to misidentification, especially when writers are proficient in multiple languages (*Markov, Nastase & Strapparava, 2020*). Furthermore, the absence of balanced and comprehensive benchmark corpora has impeded the comparison of results across studies, making it difficult to establish standardized evaluation metrics (*Cimino & Dell'Orletta, 2017*). The challenge of identifying native language interference—where aspects of a writer's first language influence their second language production—adds another layer of complexity to the NLI task (*Markov, Nastase & Strapparava, 2020*). Additionally, the rapid evolution of language in digital communication, particularly within multilingual settings, presents further challenges for traditional NLI approaches, necessitating continuous research and adaptation of methodologies to align with emerging linguistic trends (*Dey et al., 2024*).

# MATERIALS AND METHOD

## Dataset

### Data description

The NLI-PT dataset (*Río Gayo, Zampieri & Malmasi, 2018*) is the first *corpus* specifically developed for native language identification (NLI) in Portuguese. It aims to identify an author's native language based on their writing in European Portuguese as a second language. The dataset comprises 1,868 student essays authored by learners whose native languages span a diverse set, including Chinese, English, Spanish, German, Russian, French, Japanese, Italian, Dutch, Tetum, Arabic, Polish, Korean, Romanian, and Swedish. These texts were compiled from three Portuguese learner corpora: (i) COPLE2, (ii) the Leiria *corpus*, and (iii) PEAPL27, as summarized in Table 1.

NLI-PT includes the original student texts accompanied by four types of linguistic annotations: part-of-speech (POS) tags, fine-grained POS, constituency parses, and dependency parses. The three corpora consist of written texts from Portuguese learners with varying proficiency levels and different native languages. The dataset incorporates all the data from COPLE2, along with selected portions of PEAPL2 and the Leiria *corpus*. Designed for both NLI research and broader studies in Second Language Acquisition and educational NLP, NLI-PT serves as a valuable resource for various applications, such as grammatical error correction and language learning tools. The dataset is made freely available for research purposes, promoting further exploration in the field of Portuguese language learning and processing.

To gain an overview of the data, we conducted a deeper analysis of the quantity of data for each language type as well as the distribution of sample lengths. This is an important data processing step to develop appropriate approaches and effectively mine the data. The distribution of samples in each class is shown in Table 2. The amount of data across classes is relatively imbalanced, with the number of samples for Arabic, Tetum, and Swedish being relatively low, with fewer than 20 samples each, while languages such as English, Spanish, and Italian have a larger volume of over 200 samples.

The length of the text is also a significant factor affecting the accuracy of the model. To gain insights into the distribution of texts based on word length, we created a histogram

**Table 1 Dataset statistics for each source *corpus*, including the number of texts, tokens, types, and the type-to-token ratio (TTR).**

| Corpus | Texts | Tokens | Types | TTR |
|---|---|---|---|---|
| COPLE2 | 1,058 | 201,921 | 9,373 | 0.05 |
| Leiria | 330 | 57,358 | 4,504 | 0.08 |
| PEAPL2 | 480 | 121,138 | 6,808 | 0.06 |
| **Total** | **1,868** | **380,417** | **20,685** | **0.05** |

**Table 2 Number of samples per native language class.**

| Language | COPLE | PEAPL | LEIRIA | Total |
|---|---|---|---|---|
| Arabic | 13 | 1 | 0 | 14 |
| Chinese | 323 | 32 | 0 | 355 |
| Dutch | 17 | 26 | 0 | 43 |
| English | 142 | 62 | 31 | 235 |
| French | 59 | 38 | 7 | 104 |
| German | 86 | 88 | 40 | 214 |
| Italian | 49 | 83 | 83 | 215 |
| Japanese | 52 | 15 | 0 | 67 |
| Korean | 9 | 9 | 48 | 66 |
| Polish | 31 | 28 | 12 | 71 |
| Romanian | 12 | 16 | 51 | 79 |
| Russian | 80 | 11 | 1 | 92 |
| Spanish | 147 | 68 | 56 | 271 |
| Swedish | 16 | 2 | 1 | 19 |
| Tetum | 22 | 1 | 0 | 23 |
| **Total** | **1,058** | **480** | **330** | **1,868** |

categorizing them as presented in Fig. 1. Generally, most length of texts fall within the range of about 450 to 1,600.

### Data processing

In this study, we employed a data processing pipeline to prepare Portuguese text for input into the GPT-2 model. The data preprocessing phase involved several key steps to enhance the quality of the input for the GPT-2 model. Initially, we performed text normalization, which included converting all text to lowercase and removing any extraneous punctuation. Subsequently, we applied tokenization to segment the documents into individual words and phrases, facilitating more efficient processing. The dataset also includes various annotations, such as part-of-speech (POS) tagging and constituency parses, which we leveraged to enrich the input features. Furthermore, we ensured that the training data was balanced across different native language backgrounds to mitigate any biases during model training. This comprehensive data processing pipeline aimed to optimize the performance of GPT-2 in generating coherent and contextually relevant responses in Portuguese.

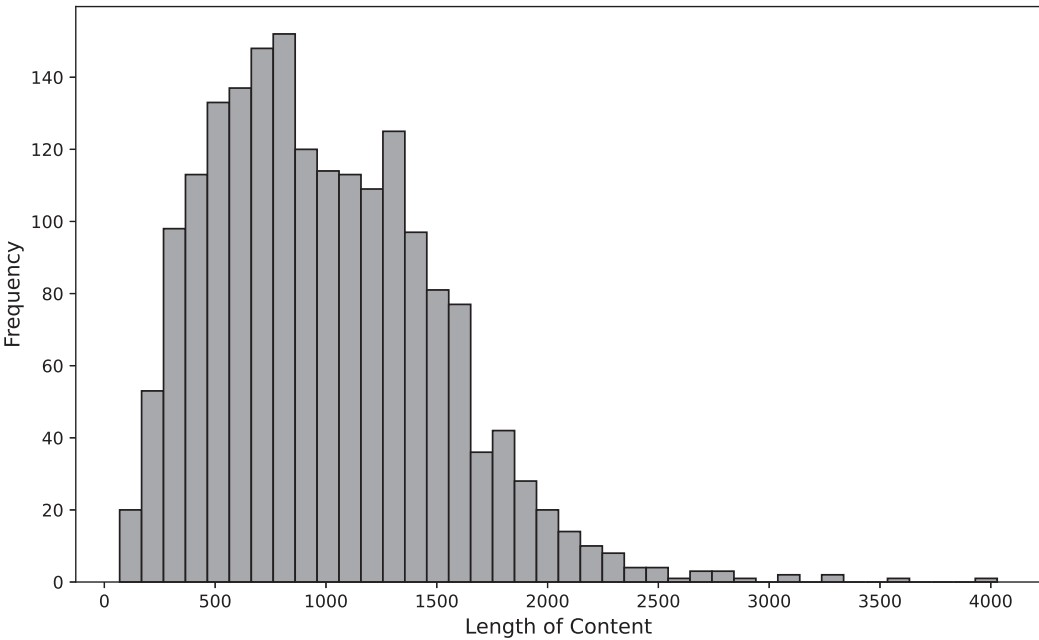

**Figure 1 Distribution of sentence lengths within the dataset.**

Next, the dataset was split into two subsets using a 90:10 ratio for training and testing, respectively, through stratified sampling. This ensured that each class was proportionally represented in both sets, providing the model with sufficient data for learning while enabling evaluation on previously unseen texts.

## Method

### Overview

In this study, we propose the framework outlined in Fig. 2 for the problem of NLI. The framework is organized into two main modules: the Finetuning Module and the Application Module.

Initially, we begin with the NLI_PT dataset, which is partitioned into two subsets: the test dataset and the finetuning dataset. For the Finetuning Module, the finetuning dataset undergoes a series of preprocessing steps, including data cleaning and processing. Subsequently, a GPT-2 model is finetuned on this dataset, resulting in the development of a pretrained GPT-2 model specifically designed for NLI.

Following this, the Application Module leverages the test dataset to predict the native language of samples using the finetuned GPT-2 model. The output from this module consists of the predicted native language for each input sample. Detailed descriptions of each processing step will be elaborated upon in the subsequent subsections.

### Proposed architecture

This section presents our approach and architectural design for identifying native languages using the GPT-2 model. Our methodology enhances robustness and accuracy

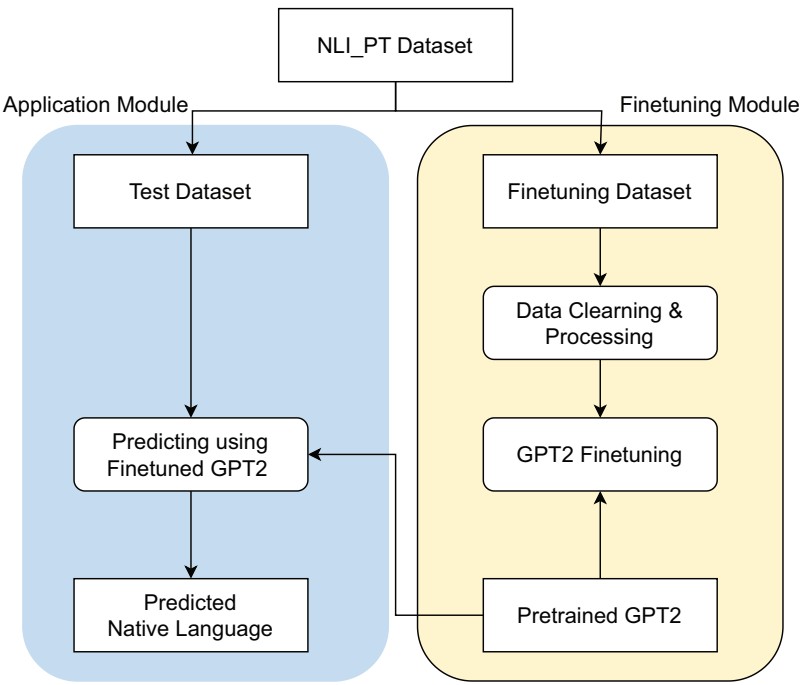

**Figure 2 Proposed framework overview.**

through an innovative embedding extraction and fusion process as presented in Fig. 3. The proposed architecture includes several integral components focused on improving the precision of NLI. Initially, we transform text documents into tokens, which are then embedded and input into the GPT-2 model. To enhance generalization, we extract embedding features from three last depth layers of the model.

NLI_PT dataset is defined as $S = \{s_j, y_j\}_{j=1}^{M}$, where $s_j$ represents an individual data point (text document) and $y_j$ denotes its corresponding label (native language), and M is number of sample in the dataset. The dataset is divided into training and testing subsets, $S_{\text{train}}$ and $S_{\text{test}}$.

During training, we utilize the embedding extraction function to derive semantic embeddings from a batch of data points $\{s_j, y_j\}_{j=1}^{b}$ in $S_{\text{train}}$, where $b$ is the batch size.

To create a unified representation for each data point, we combine all available embeddings by using concatenation, yielding the final semantic embedding $\Gamma_{(j)}$ for the $j$-th data point:

$$\Gamma_{(j)} = \begin{bmatrix} \eta_j^{(10)} \\ \eta_j^{(11)} \\ \eta_j^{(12)} \end{bmatrix}, \tag{1}$$

$$\eta_j^{(l)} = e(s_j \mid l). \tag{2}$$

Here, $l$ ranges from 10 to 12 for the GPT-2 model corresponding to the outputs of the last three transformer blocks in the architecture.

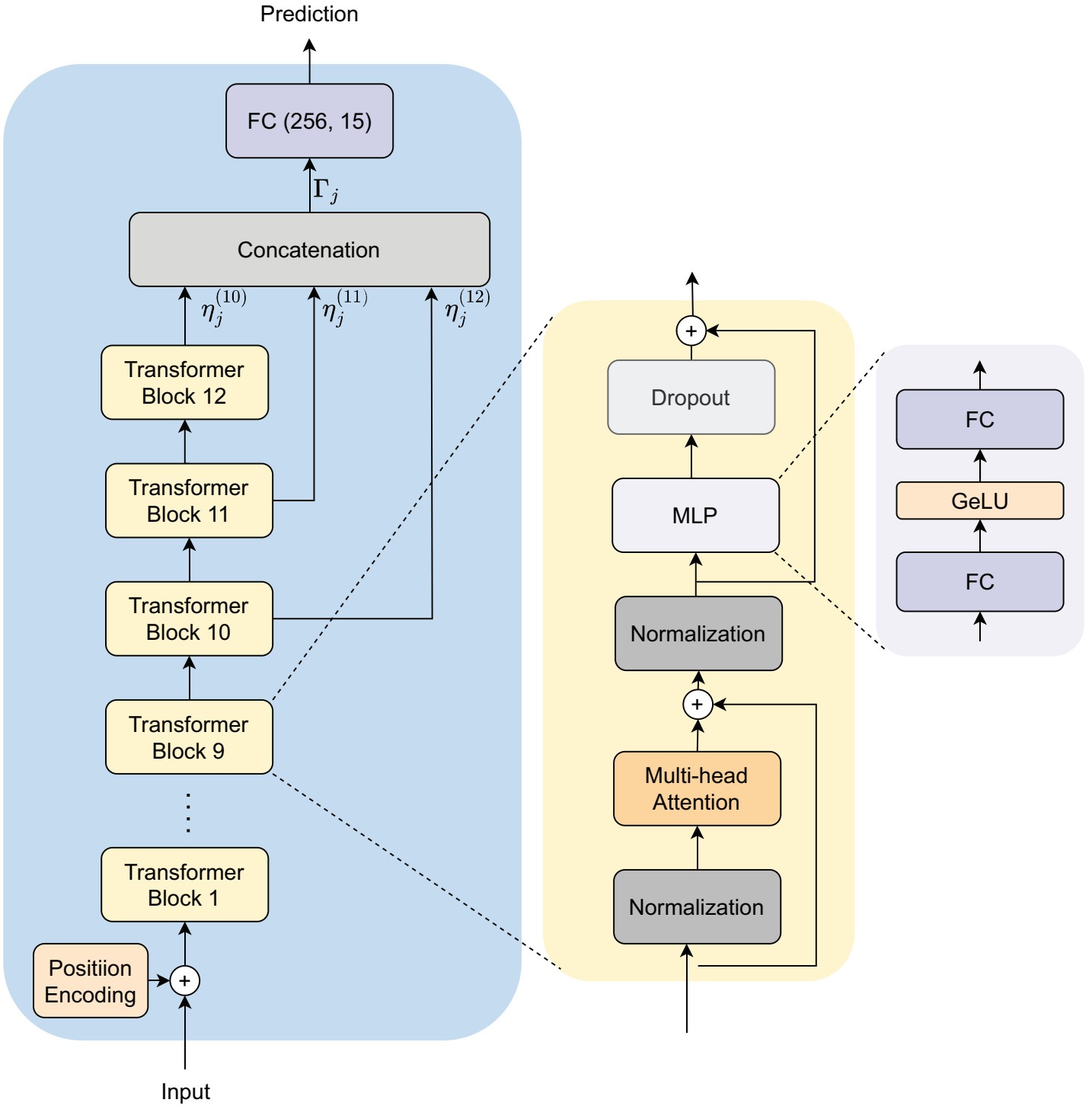

**Figure 3** **Model architecture for native language identification.**

The concatenated embeddings $\Gamma_{(j)}$ are then input into the classifier head that is a fully connected layer to predict the native language of the input sample.

### Loss function and training details

Cross-entropy loss (CEL) is a commonly used metric for multi-class classification problems. For a model predicting $K$ classes, the cross-entropy loss is defined as:

$$\text{Loss} = -\sum_{i=1}^{K} y_i \log(\hat{y}_i), \tag{3}$$

where $y_i$ is the label of class $i$ (1 if class $i$ is the true class, 0 otherwise) and $\hat{y}_i$ is the predicted probability of the model for class $i$.

We fine-tuned the model for five epochs with a learning rate of $10^{-4}$, using the Adam optimizer (*Kingma & Ba, 2014*) for efficient gradient-based optimization. The entire model and experiments were deployed on a 3060 GPU with 12 GB of memory.

## Evaluation

To evaluate the effectiveness of our fine-tuned GPT-2 embeddings for the task of NLI, we compared the performance of our method against established baseline models, including various deep learning architectures such as deep neural networks (DNNs) (*Schröder & Niekler, 2020*), convolutional neural networks (CNNs) (*Lai et al., 2015*), recurrent neural networks (RNNs) (*Liu, Qiu & Huang, 2016*), and LSTM (Long Short-Term Memory networks) (*Nowak, Taspinar & Scherer, 2017*). Additionally, we will incorporate traditional machine learning techniques such as support vector machine (SVM) and RF (Random Forest) from extracted features of Word2Vec (*Mikolov et al., 2013*). To evaluate the performance of those models, we calculated various metrics, including F1-score, recall, precision and accuracy. Our objective is to assess whether the fine-tuned GPT-2 model significant improvements in accurately identifying a speaker's native language compared to the performances of the baseline models.

To conduct a comprehensive performance analysis, we compared with powerful pre-trained models for language processing tasks. These models include:

- **Bidirectional encoder representations from transformers (BERT)** (*Devlin et al., 2018*) is a powerful language model based on the Transformer architecture, capable of generating deep bidirectional semantic representations by capturing the context of words in both forward and backward directions. This bidirectional understanding is particularly valuable for language classification tasks. BERT has been successfully applied in a wide range of domains (*Leow, Nguyen & Chua, 2021*; *Nguyen-Vo et al., 2021*; *Nguyen et al., 2024*), demonstrating its versatility and effectiveness.
- **SciBERT** (*Beltagy, Lo & Cohan, 2019*): Designed specifically for scientific text, SciBERT leverages a large *corpus* of scientific literature to improve performance on domain-specific tasks, making it particularly effective for identifying languages used in scientific contexts.

**Table 3 Model architectures and default parameter settings used in the experiments.**

| Model | | Architecture type | Parameter settings (Default) |
|---|---|---|---|
| Baseline | DNN | Fully Connected Neural Network | 4 hidden layers, dropout = 0.25 |
| | CNN | 1D Convolutional Neural Network | 3 1DCNN layers, 1 MaxPooling, |
| | RNN | Recurrent Neural Network | 2 layers, 128 hidden units |
| | LSTM | Long Short-Term Memory | 2 layers, 256 hidden units, |
| | Word2vec + SVM | Word Embedding + ML | Linear kernel, C = 1.0 |
| | Word2vec + RF | | 100 trees, max_depth = 20 |
| Pretrained | BERT | Based Transformer | 12 layers, 12 attention heads |
| | SciBERT | | |
| | BioBERT | | |
| | BlueBERT | | |
| | RoBERTa | | |

- **BioBERT** (*Lee et al., 2019*): This model extends BERT for biomedical text mining by pre-training on large-scale biomedical corpora, enhancing its ability to identify languages in medical and biological documents.
- **BlueBERT** (*Peng, Yan & Lu, 2019*): A model optimized for biomedical and clinical text, BlueBERT incorporates knowledge from both general and biomedical corpora, providing robust performance in language identification across diverse biomedical datasets.
- **RoBERTa** (*Liu et al., 2019*): An optimized version of BERT, RoBERTa improves performance by carefully tuning training parameters and using a larger dataset, thereby enhancing its capacity to identify languages in various texts effectively.

The model architectures and their corresponding default parameter settings used in the experiments are summarize in Table 3. Through a comparison of proposed model with these methods, we aim to showcase the effectiveness of our architecture, especially in improving semantic understanding, feature extraction techniques, and addressing the intricate characteristics of NLI.

# EXPERIMENTAL RESULTS AND DISCUSSION

## Performance analysis with baseline models

Table 4 provides a comprehensive comparison between the proposed GPT-2 model and a range of baseline models, including traditional deep learning architectures (DNN, CNN, RNN, long short-term memory (LSTM)) and hybrid models using Word2Vec embeddings combined with SVM or RF classifiers.

Among all models evaluated, GPT-2 demonstrates a clear and consistent superiority across all performance metrics. It achieves the highest scores in Weighted F1 (0.9419), Macro F1 (0.8521), and Micro F1 (0.9465), indicating both class-wise balance and overall accuracy. Similarly, GPT-2 leads in recall metrics, including Weighted Recall (0.9465), Macro Recall (0.8534), and Micro Recall (0.9465), showcasing its ability to capture true

**Table 4 Performance comparison with baseline models.** Bold denotes the best-performing value for each metric.

| Baseline model | F1 | | Recall | | Precision | | Accuracy |
|---|---|---|---|---|---|---|---|
| DNN | Weighted F1 | 0.6672 | Weighted Recall | 0.7005 | Weighted Precision | 0.7094 | 0.7005 |
| | Marco F1 | 0.5157 | Marco Recall | 0.5044 | Marco Precision | 0.6234 | |
| | Micro F1 | 0.7005 | Micro Recall | 0.7005 | Micro Precision | 0.7005 | |
| CNN | Weighted F1 | 0.6985 | Weighted Recall | 0.7219 | Weighted Precision | 0.7538 | 0.7219 |
| | Marco F1 | 0.5376 | Marco Recall | 0.5362 | Marco Precision | 0.6460 | |
| | Micro F1 | 0.7219 | Micro Recall | 0.7219 | Micro Precision | 0.7219 | |
| RNN | Weighted F1 | 0.7381 | Weighted Recall | 0.7594 | Weighted Precision | 0.7449 | 0.7594 |
| | Marco F1 | 0.5512 | Marco Recall | 0.5440 | Marco Precision | 0.6077 | |
| | Micro F1 | 0.7594 | Micro Recall | 0.7594 | Micro Precision | 0.7594 | |
| LSTM | Weighted F1 | 0.7499 | Weighted Recall | 0.7701 | Weighted Precision | 0.7863 | 0.7701 |
| | Marco F1 | 0.5800 | Marco Recall | 0.5583 | Marco Precision | 0.6805 | |
| | Micro F1 | 0.7701 | Micro Recall | 0.7701 | Micro Precision | 0.7701 | |
| Word2vec + SVM | Weighted F1 | 0.7722 | Weighted Recall | 0.8021 | Weighted Precision | 0.7705 | 0.8021 |
| | Marco F1 | 0.5821 | Marco Recall | 0.5915 | Marco Precision | 0.5963 | |
| | Micro F1 | 0.8021 | Micro Recall | 0.8021 | Micro Precision | 0.8021 | |
| Word2vec + RF | Weighted F1 | 0.8115 | Weighted Recall | 0.8342 | Weighted Precision | 0.8256 | 0.8342 |
| | Marco F1 | 0.6460 | Marco Recall | 0.6297 | Marco Precision | 0.7098 | |
| | Micro F1 | 0.8342 | Micro Recall | 0.8342 | Micro Precision | 0.8342 | |
| GPT-2 | Weighted F1 | **0.9419** | Weighted Recall | **0.9465** | Weighted Precision | **0.9459** | **0.9465** |
| | Marco F1 | **0.8521** | Marco Recall | **0.8534** | Marco Precision | **0.8779** | |
| | Micro F1 | **0.9465** | Micro Recall | **0.9465** | Micro Precision | **0.9465** | |

positives effectively across both majority and minority classes. Its precision is equally strong, with the highest Weighted Precision (0.9459), Macro Precision (0.8779), and Micro Precision (0.9465), underscoring the model's robustness in minimizing false positives.

Compared to the strongest baseline, Word2Vec + RF, which achieves a Weighted F1 of 0.8115 and an accuracy of 0.8342, GPT-2 shows a significant performance gain of over 13% in F1 and more than 11% in accuracy. Deep learning models like LSTM and RNN perform moderately well, with LSTM achieving a Weighted F1 of 0.7499 and accuracy of 0.7701, but still fall short in macro-level metrics, indicating limited ability to generalize across less-represented classes.

The consistent outperformance of GPT-2 can be attributed to its large-scale pretraining on diverse textual corpora and its attention-based architecture, which allows it to capture nuanced contextual dependencies. Unlike traditional baselines that rely on handcrafted features or fixed embeddings, GPT-2 dynamically generates contextualized representations that adapt to the linguistic features of each input, contributing to its strong generalization capability.

These results highlight the value of transformer-based architectures, particularly in complex classification tasks such as native language identification, where class imbalance and linguistic diversity pose significant challenges. The superior performance of GPT-2

**Table 5 Performance comparison with pre-trained models.** Bold denotes the best-performing value for each metric.

| Pretrained models | F1 | | Recall | | Precision | | Accuracy |
|---|---|---|---|---|---|---|---|
| BERT | Weighted F1 | 0.8817 | Weighted Recall | 0.8877 | Weighted Precision | 0.8958 | 0.8877 |
| | Macro F1 | 0.7760 | Macro Recall | 0.7495 | Macro Precision | 0.8406 | |
| | Micro F1 | 0.8877 | Micro Recall | 0.8877 | Micro Precision | 0.8877 | |
| RoBERTa | Weighted F1 | 0.8994 | Weighted Recall | 0.9144 | Weighted Precision | 0.8913 | 0.9144 |
| | Macro F1 | 0.7314 | Macro Recall | 0.7252 | Macro Precision | 0.7500 | |
| | Micro F1 | 0.9144 | Micro Recall | 0.9144 | Micro Precision | 0.9144 | |
| SciBERT | Weighted F1 | 0.9029 | Weighted Recall | 0.9144 | Weighted Precision | 0.9046 | 0.9144 |
| | Macro F1 | 0.7680 | Macro Recall | 0.7651 | Macro Precision | 0.7939 | |
| | Micro F1 | 0.9144 | Micro Recall | 0.9144 | Micro Precision | 0.9144 | |
| BioBERT | Weighted F1 | 0.8817 | Weighted Recall | 0.8930 | Weighted Precision | 0.8903 | 0.8930 |
| | Macro F1 | 0.7945 | Macro Recall | 0.7830 | Macro Precision | 0.8496 | |
| | Micro F1 | 0.8930 | Micro Recall | 0.8930 | Micro Precision | 0.8930 | |
| BlueBERT | Weighted F1 | 0.8981 | Weighted Recall | 0.9037 | Weighted Precision | 0.9084 | 0.9037 |
| | Macro F1 | 0.8139 | Macro Recall | 0.8010 | Macro Precision | 0.8605 | |
| | Micro F1 | 0.9037 | Micro Recall | 0.9037 | Micro Precision | 0.9037 | |
| GPT-2 | Weighted F1 | **0.9419** | Weighted Recall | **0.9465** | Weighted Precision | **0.9459** | **0.9465** |
| | Macro F1 | **0.8521** | Macro Recall | **0.8534** | Macro Precision | **0.8779** | |
| | Micro F1 | **0.9465** | Micro Recall | **0.9465** | Micro Precision | **0.9465** | |

demonstrates its suitability for real-world applications requiring both accuracy and adaptability across varied language backgrounds.

## Performance analysis with pre-trained models

The comparative analysis of pre-trained models, as presented in Table 5, highlights significant differences in performance across multiple evaluation metrics. Among the models evaluated, the fine-tuned GPT-2 consistently outperformed both general-purpose transformer models, such as BERT and RoBERTa, and domain-specific variants, including BioBERT and BlueBERT. Notably, GPT-2 achieved the highest scores across key metrics–Weighted F1 (0.9419), Macro F1 (0.8521), and accuracy (0.9465)–demonstrating superior robustness and generalization across diverse class distributions.

While models such as SciBERT and RoBERTa exhibited strong micro-level performance—indicating their effectiveness in predicting frequent classes—the fine-tuned GPT-2 model distinguished itself by delivering balanced improvements across all evaluation metrics, including macro-level scores. This balance suggests that GPT-2 is not only adept at identifying dominant classes but also demonstrates strong capability in handling minority categories with precision. This is further evidenced by its high Macro Recall (0.8534) and Macro Precision (0.8779), which underscore the model's ability to maintain consistent performance despite class imbalance.

In contrast, domain-specific models like BioBERT and BlueBERT, although optimized for biomedical text, performed slightly below general-purpose transformers such as

**Table 6 Performance comparison using different combinations of the final transformer blocks.** Bold denotes the best-performing value for each metric.

| Block combination | F1 | | Recall | | Precision | | Accuracy |
|---|---|---|---|---|---|---|---|
| Last 1 Block (Layer 12) | Weighted F1 | 0.9182 | Weighted Recall | 0.9210 | Weighted Precision | 0.9201 | 0.9210 |
| | Macro F1 | 0.8284 | Macro Recall | 0.8297 | Macro Precision | 0.8502 | |
| | Micro F1 | 0.9210 | Micro Recall | 0.9210 | Micro Precision | 0.9210 | |
| Last 2 Blocks (Layers 11–12) | Weighted F1 | 0.9314 | Weighted Recall | 0.9352 | Weighted Precision | 0.9335 | 0.9352 |
| | Macro F1 | 0.8426 | Macro Recall | 0.8437 | Macro Precision | 0.8668 | |
| | Micro F1 | 0.9352 | Micro Recall | 0.9352 | Micro Precision | 0.9352 | |
| Last 3 Blocks (Layers 10–12) | Weighted F1 | **0.9419** | Weighted Recall | **0.9465** | Weighted Precision | **0.9459** | **0.9465** |
| | Macro F1 | **0.8521** | Macro Recall | **0.8534** | Macro Precision | **0.8779** | |
| | Micro F1 | **0.9465** | Micro Recall | **0.9465** | Micro Precision | **0.9465** | |

RoBERTa. This may reflect a domain mismatch, where the models' specialized training fails to confer a performance advantage in broader language classification tasks.

The consistently strong results of GPT-2 can be attributed to its large-scale pretraining on diverse corpora and the use of advanced attention-based mechanisms, which enable a deeper contextual understanding of language. These characteristics make GPT-2 particularly well-suited for complex classification scenarios. Overall, the findings highlight the importance of leveraging modern transformer architectures—particularly those with broad pretraining and high contextual capacity—for achieving state-of-the-art results in native language identification and related tasks.

## Effect of transformer block selection

To assess the impact of different transformer block combinations on embedding quality, we conducted a comparative experiment using three configurations of the GPT-2 model: (i) the final block (Layer 12), (ii) the last two blocks (Layers 11–12), and (iii) the last three blocks (Layers 10–12). This experiment aimed to validate the design decision to utilize the final three layers for embedding extraction, as implemented in our proposed architecture. For each configuration, the output embeddings from the selected layers were concatenated and passed through an identical classification head. To ensure a fair comparison, all other experimental conditions—including the optimizer, learning rate, and batch size—were held constant across the experiments.

The results presented in Table 6 indicate that utilizing a greater number of transformer blocks for semantic embedding extraction leads to improved model performance across all evaluation metrics. Specifically, the configuration incorporating the final three blocks (Layers 10–12) achieves the highest accuracy (94.65%) and F1 scores, suggesting that aggregating information from multiple layers produces more informative and discriminative semantic representations. In contrast, relying solely on the final block results in noticeably lower performance, likely due to the reduced contextual depth captured. This trend underscores the value of deeper fusion of transformer layer outputs in enhancing generalization and semantic sensitivity for the native language identification

task. Accordingly, the choice to employ the final three layers in the proposed architecture is empirically validated.

## LIMITATIONS AND FUTURE WORK

While our approach to NLI using a fine-tuned GPT-2 model has demonstrated strong performance, several limitations and future directions warrant consideration to enhance its applicability and generalizability.

First, the NLI-PT dataset used in this study comprises learner texts from a limited set of linguistic backgrounds, which may restrict the generalizability of our model across more diverse populations. Learners with underrepresented native languages, dialects, or multilingual profiles may exhibit language transfer effects not captured in the current dataset. Future work should incorporate more heterogeneous corpora encompassing broader linguistic, cultural, and educational contexts to foster inclusivity and improve the robustness of the model.

Second, the reliance on large-scale transformer models like GPT-2 poses practical challenges in terms of computational resource requirements. This can hinder real-world deployment, especially in low-resource environments or educational institutions lacking access to high-performance computing. To address this, future research could leverage lightweight and efficient fine-tuning strategies such as low-rank adaptation (LoRA), which has shown promise in reducing training costs without sacrificing performance (*Chen et al., 2024*). Additionally, emerging techniques such as integrating reasoning frameworks like Tree of Thoughts and behavior-driven adaptation (*Ding et al., 2023*; *Wang et al., 2025*) may help tailor transformer models to specific user profiles or contexts with minimal overhead.

By extending this work toward more inclusive datasets and efficient fine-tuning methods, future research can advance the scalability and practical deployment of transformer-based NLI systems, ultimately supporting broader adoption in diverse linguistic and educational settings.

## CONCLUSIONS

This study explored the application of a fine-tuned GPT-2 model for NLI using the NLI-PT dataset, demonstrating the potential of transformer-based architectures in accurately identifying learners' native languages from their second-language writing. Our model outperformed traditional machine learning approaches and other pre-trained language models in terms of accuracy, F1-score, and recall, highlighting the effectiveness of deep contextual representations in handling complex linguistic patterns.

By leveraging the strengths of GPT-2, including its bidirectional understanding and capacity for modeling complex language structures, we were able to address key challenges in NLI, such as ambiguity in learner language and variability in writing styles. The results underscore the value of large-scale language models in advancing NLI research and supporting applications in personalized education, forensic linguistics, and multilingual language technologies.

Overall, this work provides a solid foundation for future studies aiming to enhance NLI systems through transformer-based models. Future research should aim to extend this framework to more diverse datasets and explore efficient deployment strategies to ensure broader accessibility and scalability across real-world applications.

### Funding
The authors received no funding for this work.

### Competing Interests
The authors declare that they have no competing interests.

### Author Contributions
- Yuzhe Nie conceived and designed the experiments, performed the experiments, analyzed the data, performed the computation work, prepared figures and/or tables, authored or reviewed drafts of the article, and approved the final draft.

### Data Availability
The code and data used in the experiments are available in the Supplemental Files.

The dataset is originally from: A Portuguese Native Language Identification Dataset, https://aclanthology.org/W18-0534.

### Supplemental Information
Supplemental information for this article can be found online at http://dx.doi.org/10.7717/peerj-cs.2909#supplemental-information.

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
