# Peer review of "Native language identification from text using a fine-tuned GPT-2 model"

_PeerJ Computer Science, doi:10.7717/peerj-cs.2909_

## Round 0.1 · original submission · Major Revisions

Please incorporate the reviewers' feedback and make the necessary revisions to the manuscript.

Reviewer 1 ·

Basic reporting

- The paper is mostly well written and easy to follow. The background and related research are clearly explained. Figures and tables are clear and help explain the study well. However, there are a few points the authors should clarify.

Experimental design

- It is good that the authors provided the source code and dataset.
- The proposed method is explained clearly, and the diagrams help to show how the model works.
- The data processing steps and evaluation metrics are also explained in a simple and clear way.

However, a few things could make the paper stronger:
- It would help to include a feature analysis using methods like t-SNE or PCA. This would show how well the pretrained models are learning patterns in the data.
- The sentence: “Next, the dataset was divided into three parts using a 90:10 ratio for training and testing” is confusing. It is unclear how it was actually split. Please clarify and include the exact number of samples used for training and testing to make this more transparent.

Validity of the findings

- The paper would be clearer if it gave more detail about how the models were set up—especially the baseline and pretrained versions. Please include information like architecture type, parameter settings, and training details. This would help others reproduce the work and properly judge the experimental setup.

Additional comments

- There seems to be a formatting issue in the table near line 160. It would be good to fix this so everything is consistent and easy to read.

Cite this review as

Reviewer 2 ·

Basic reporting

+ The manuscript is generally well-organized and written in readable English. However, the motivation currently embedded within the Introduction lacks clarity and depth. The authors are encouraged to restructure this part by creating a dedicated section titled “Motivation” to better highlight the research problem and its significance.
+ The repeated full-form usage of terms such as Native Language Identification (NLI) throughout the paper reduces conciseness. The authors are advised to define abbreviations upon first mention and use abbreviations thereafter to maintain a streamlined narrative.

Experimental design

+ The dataset section would benefit from clearer structural organization. Specifically, the content between lines 155–161 should be reallocated under a separate subsection titled Data Description, as it does not align well with the surrounding context.
+ The rationale for using outputs from the last three convolutional blocks in pretrained models remains unexplained. A comparative experiment evaluating the use of different block combinations (e.g., last 1, 2, or 3 blocks) should be conducted to justify the chosen configuration.

Validity of the findings

+ The subsection Performance Analysis with Pretrained Models presents an overly fragmented and verbose discussion. The authors should consolidate the visualizations into a single comparative table, similar to Table 3, to better support cross-model comparison.
+ A dedicated section for feature representation visualization across the baseline and pretrained models would add significant insight into how different models encode linguistic or visual cues.
+ The manuscript should include a subsection outlining future research directions, particularly in terms of scaling the methodology, applying to multilingual data, or adapting to low-resource settings.

Additional comments

+ Please double-check and revise the mapping table formatting on Line 160 to improve readability and correctness.

Cite this review as

---

## Round 0.2 · accepted · Accept

The authors have addressed all reviewers' comments. Based on their recommendations and my own assessment, I recommend accepting this manuscript for publication.

Reviewer 1 ·

Basic reporting

- The manuscript is clearly written and well-structured. Figures and tables are clear and effectively support the proposal method
- Background and related work are appropriately cited and provide solid context.
- Previous issues have been addressed in the revision.

Experimental design

- The use of the NLI-PT dataset is appropriate and adequately described.
- The methodology is well-explained, with clear details on data processing and model architecture.
- Additional experiments were conducted to support the model configuration, improving the study.

Validity of the findings

- Evaluation metrics and comparisons are comprehensive and support the study’s conclusions.
- Details on model parameters and training configurations were added for transparency.
- Reviewer suggestions on visualization and structure were addressed with clear justifications.

Cite this review as

Reviewer 2 ·

Basic reporting

The revised version is well-organized and clearly written. I don't have any other comments.

Experimental design

I have no further comments.

Validity of the findings

I have no further comments.

Additional comments

No further comments.

Cite this review as